# Oxytocin Downregulates the Ca_V_1.2 L-Type Ca^2+^ Channel via Gi/cAMP/PKA/CREB Signaling Pathway in Cardiomyocytes

**DOI:** 10.3390/membranes11040234

**Published:** 2021-03-25

**Authors:** Masaki Morishima, Shintaro Tahara, Yan Wang, Katsushige Ono

**Affiliations:** 1Department of Pathophysiology, Oita University School of Medicine, Oita 879-5593, Japan; mmoris@nara.kindai.ac.jp (M.M.); heart222heart222@gmail.com (S.T.); wang@oita-u.ac.jp (Y.W.); 2Department of Food Science and Nutrition, Kindai University Faculty of Agriculture, Nara 631-8505, Japan

**Keywords:** oxytocin, L-type Ca^2+^ channel, gene transcription, CREB

## Abstract

Oxytocin (OT) and its receptor (OTR) are expressed in the heart and are involved in the physiological cardiovascular functional system. Although it is known that OT/OTR signaling is cardioprotective by reducing the inflammatory response and improving cardiovascular function, the role of OT in the cardiac electrical excitation modulation has not been clarified. This study investigates the molecular mechanism of the action of OT on cardiomyocyte membrane excitation focusing on the L-type Ca^2+^ channel. Our methodology uses molecular biological methods and a patch-clamp technique on rat cardiomyocytes with OT, combined with several signal inhibitors and/or activators. Our results show that long-term treatment of OT significantly decreases the expression of Cav1.2 mRNA, and reduces the L-type Ca^2+^ channel current (I_Ca.L_) in cardiomyocytes. OT downregulates the phosphorylated component of a transcription factor adenosine-3′,5′-cyclic monophosphate (cAMP) response element binding protein (CREB), whose action is blocked by OTR antagonist and pertussis toxin, a specific inhibitor of the inhibitory GTP-binding regulators of adenylate cyclase, Gi. On the other hand, the upregulation of Cav1.2 mRNA expression by isoproterenol is halted by OT. Furthermore, inhibition of phospholipase C (PLC) was without effect on the OT action to downregulate Cav1.2 mRNA—which suggests a signal pathway of Gi/protein kinase A (PKA)/CREB mediated by OT/OTR. These findings indicate novel signaling pathways of OT contributing to a downregulation of the Cav1.2-L-type Ca^2+^ channel in cardiomyocytes.

## 1. Introduction

Oxytocin (OT) is a neurohypophyseal hormone secreted from both central and peripheral tissues, resulting in reproduction, brain neuromodulation, and central regulation of blood pressure [1]. Although OT is commonly known as a female hormone, besides reproductive properties, there is growing evidence that OT plays a role in regulating cardiovascular function. OT acts upon the OT receptor (OTR), which belongs to the family of G-protein-coupled receptor (GPCRs). These receptors are abundantly expressed in the heart, and local OT action in the heart is important for blood pressure regulation [2], as well as cardioprotection associated with secretion of atrial natriuretic peptide (ANP) and production of nitric oxide (NO) [3,4]. Similarly, OT is associated with a direct, centrally mediated regulation of autonomic outflow to the heart leading to negative inotropic and chronotropic effects [5]. Under the pathophysiological condition, OT has anti-inflammatory and anti-fibrotic effects that could explain the observed myocardial protection in in vivo studies [6]. Recent evidence indicates that OT stimulates the proliferation and differentiation of embryonic stem cells to cardiomyocytes [7]. Although many of these studies demonstrated that OT reduced the heart rate and the force of atrial contractions [1], the potential role of oxytocin in cardiac electrical excitation has not been intensively investigated. 

Changes in ion channel expression modulation are hallmarks of heart failure-related electrical remodeling. Heart failure is defined as the end-stage of various cardiac diseases, including ischemic or dilated cardiomyopathy [8]. Chronic stimulation of β-adrenergic receptors by catecholamines leading to activation of cAMP-dependent signal transduction pathway plays a central role in the pathogenesis in heart failure [9]. Accordingly, it is postulated that transcription factors that are related to catecholamine-dependent cardiac complications, including CREB, play an important mechanism for catecholamine-dependent gene controls and altered-gene regulation in heart failure. The voltage-gated L-type Ca^2+^ channel is one of the ion channels responsible for heart failure-associated electrical remodeling in cardiomyocytes. Previous studies have demonstrated that the positive inotropic effect of noradrenaline to increase Ca^2+^ influx through the L-type Ca^2+^ channel current (I_Ca.L_) was accelerated under the prolonged intensive stimulation of the β-adrenergic receptor in the heart [10]. Although it is well known that cardiac L-type Ca^2+^ channels function is modulated by a cAMP-dependent protein kinase, little is known about the molecular mechanisms regulating L-type Ca^2+^ channel gene expression in cardiomyocytes. Although we have previously demonstrated that OT downregulated the Cav1.2-L-type Ca^2+^ channel expression accompanied by a reduction of I_Ca.L_ in cardiomyocytes [11], the signal pathways underlying cardiac OT actions are poorly understood. 

In the present study, we investigated the intracellular molecular mechanism for L-type Ca^2+^ channel expression regulation by OT. We revealed a novel function of OT/OTR signals which act in concert to regulate the transcription of the Cav1.2 channel via the reduction of distinct downstream effectors, Gi/PKA/CREB pathway. These pathways in cardiomyocytes will likely be related to the rescue of the heart in pathological conditions, such as heart failure.

## 2. Results

### 2.1. OT and OTR in Cardiomyocytes 

We first examined whether OT and OTR could be detected in the adult rat heart and neonatal rat ventricular cardiomyocytes by using quantitative real-time PCR. Figure 1 clearly demonstrates that OT and OTR are detectably expressed in adult and neonatal rat hearts, in accordance with previous studies [1]. Of note, there was an approximate 3.5-fold difference of OT mRNA expression between the atrium and the ventricles in the adult heart (Figure 1A,B). Interestingly, the levels of OTR mRNA expression were significantly higher in the atrium than in the ventricle approximately by 4.5 times in the adult heart, which is nearly identical to the difference of OT mRNA between the atrium and the ventricle (Figure 1A,C). The expression levels of OT and OTR in neonatal ventricular cardiomyocytes were nearly identical to those in adult atrial tissue. These data indicate that the expression of OT and OTR is appreciably high in neonatal cardiomyocytes. It is reasonable to evaluate the action of OT by use of neonatal cardiomyocytes, which also strengthens the rationale to utilize neonatal rat ventricular cardiomyocytes to access the effect of OT for this purpose. 

### 2.2. Long-Term Effects of OT on I_Ca.L_

We decided to evaluate the long-term effects of OT on the L-type Ca^2+^ channel. A long-term treatment (24 h) of neonatal rat ventricular cardiomyocytes with OT (10^−7^ M) significantly decreased expression of Cav1.2 mRNA by 41%, which was blocked by an OTR antagonist (OTRA). These results clearly indicate that OT exerted its effect on the L-type Ca^2+^ channel via the OTR signaling pathway (Figure 2A). Importantly, L-type Ca^2+^ channel current (I_Ca.L_) was also reduced by OT. The maximum inward I_Ca.L_ was decreased by 20.8% (Figure 2B), and the maximum chord conductance (G_Ca.L_) was reduced by 18.9% by long-term treatment with OT (Figure 2C). These electrophysiological results verified the reduction of Cav1.2 by OT (Figure 2A) and our previous observations [11].

### 2.3. OT Downregulates Ca_v_1.2 mRNA and I_Ca.L_ via the Gi Protein

To investigate the signal pathway for the downregulation of Cav1.2 and I_Ca.L_ by OT, we examined the possible roles of OTR focusing on G proteins, because OTR differentially signals via Gq and Gi proteins [12]. Interestingly, OT was unable to reduce Cav1.2 mRNA when cardiomyocytes were pretreated with a specific inhibitor of the Gi protein, pertussis toxin (100 ng/mL), for 24 h. However, inhibition of Gq-PLC-induced phosphoinositide hydrolysis with a compound U73122 (2 μM) could not block the effects of OT. 

Given our results, it is suggested that OT/OTR activates the Gi protein to interfere with cellular cAMP productions, which is similar to the action of acetylcholine binding to muscarinic receptors to inhibit activation of adenylate cyclase. To confirm this hypothesis, we then applied isoproterenol, forskolin, or acetylcholine to cardiomyocytes for 24 h to observe the changes of Cav1.2 mRNA. Importantly, isoproterenol and forskolin significantly increased the expression of Cav1.2 mRNA, and acetylcholine significantly decreased the expression of Cav1.2 mRNA (Figure 3). Of note, OT significantly attenuated the increases of Cav1.2 mRNA caused by isoproterenol or forskolin, suggesting the importance of [cAMP]_I_ modulation by OT to regulate Cav1.2 transcription. Electrophysiological evaluation of I_Ca.L_ supports RT-PCR data demonstrating that OT and acetylcholine downregulated I_Ca.L_, and isoproterenol upregulated I_Ca.L_ in cardiomyocytes when applied for 24 h (Figure 3C,D). 

### 2.4. Effects of OT on Protein Kinase Activity

Multiple G protein-coupled receptors in the heart act through cAMP/PKA pathways to regulate a number of cellular proteins, including the L-type Ca^2+^ channel. To investigate whether downregulation of Cav1.2 expression by OT was dependent on PKA signaling, we measured cytosolic PKA enzyme activity in OT-treated cardiomyocytes in conjunction with several compounds that regulate β-adrenergic and cholinergic receptor functions (Figure 4). Long-term treatment with OT (10^−7^ M) significantly decreased PKA activity (−47 ± 3%) in cardiomyocytes in comparison with vehicle treatment in experiment condition A (Figure 4A). Consistent with the result in Figure 2A, OTRA completely blocked the effect of OT in downregulating PKA activity. Similarly, when cardiomyocytes were pretreated with pertussis toxin, OT was unable to reduce PKA activity. These observations tightly confirmed the action of the Gi protein as a downstream signal of OT/OTR pathway. Because assay kit enzyme activities vary upon product lot numbers, results using different PKA activity assay kits were evaluated independently as experiment A (panel A) and B (panel B). Likewise, long-term treatment with OT (10^−7^ M) significantly decreased PKA activity in cardiomyocytes (−64 ± 8%) in comparison with vehicle treatment in experiment condition B (Figure 4B). In this condition, we applied isoproterenol, forskolin, and a non-selective phosphodiesterase inhibitor 3-isobutyl-1-methylxanthine (IBMX) in the culture medium to confirm their action to increase PKA activity in cardiomyocytes as positive controls. Importantly, OT significantly attenuated the PKA activity caused by isoproterenol and forskolin. In addition, we confirmed the action of acetylcholine to reduce PKA activity as a negative/positive control. In the present study, we verified that OT reduced basal PKA activity likely through the OTR/Gi protein pathway.

### 2.5. Effects of OT on CREB Phosphorylation

Because the cAMP/PKA pathway is now clarified to regulate expression of Cav1.2 gene, and because PKA phosphorylates and consequently activates a transcription factor cyclic AMP response element binding protein (CREB), we then examined whether downregulation of Ca_v_1.2 mRNA expression by OT was associated with lower CREB phosphorylation level in the nucleus. Western blot analysis revealed that phosphorylated CREB in the nucleus was significantly decreased by OT when assessed at 24 h after OT treatment, and the reduction was even larger when the treatment was prolonged for 48 h (Figure 5A). Similarly, the result of PKA downregulation by OT through OTR in Figure 4, a decrease of phosphorylated CREB by OT was completely blocked by OTRA (Figure 5B), suggesting again that OT exerted its effects through OT/OTR pathways. Moreover, a decrease of phosphorylated CREB by OT was completely blocked by a specific inhibitor of Gi protein pertussis toxin (Figure 5C), suggesting that inactivation of PKA activity by OT is sufficient to reduce phosphorylation levels of CREB in the nucleus in this study. The role of the G-Protein-coupled receptor signaling pathway on the CREB phosphorylation was finally confirmed by three ligands, OT, isoproterenol, and acetylcholine (Figure 5D). Given all together, these results indicate that OT can modulate myocardial Ca^2+^ homeostasis via regulating sarcoplasmic membrane expression levels of the voltage-gated Ca^2+^ channel, which is qualitatively comparable with parasympathetic/acetylcholine signals in cardiomyocytes.

## 3. Discussion

Our study demonstrates that OT downregulate the Cav1.2 channel leading to a reduction of I_Ca.L_ in cardiomyocytes as a long-term effect, which is accompanied by a decrease of p-CREB in the nucleus. The reduction of Cav1.2 by OT was prevented by pertussis toxin, but not by a Gq/11 inhibitor U73112, suggesting that the signal is mediated by Gi in suppressing PKA-dependent CREB phosphorylation. Moreover, isoproterenol- or forskolin-treatment demonstrated an opposite result, whereas acetylcholine-treatment demonstrated a similar effect to that of OT on Cav1.2 expression, which indicates that OT/OTR signal pathway on the myocardial Ca^2+^ modulation is similar to those of parasympathetic stimulation. This study provides new information shedding light on OT as a cardiovascular-regulating hormone by modulating transcription of the Cav1.2 channel.

### 3.1. Myocardial Ca^2+^ Homeostasis Modulated by OT

Oxytocin, a female reproductive hormone, has well-established functions in parturition, lactation, and mental health. The present study explored the actions of OT beyond its traditionally recognize central (e.g., social behaviors) and peripheral (e.g., uterine contraction) actions. We have previously reported that OT is a cardiovascular modulator that acts primarily to downregulate Cav1.2-L-type Ca^2+^ channel, but not Cav1.3, Cav3.1, or Cav3.2 channels in cardiomyocytes [11]. In addition, OT decreased I_Ca.L_ by the long-term effect (24–72 h), but not by the short-term effect (5 min) [11] Although these long-term effects of OT on the Cav1.2 channel was obvious, the underlying mechanism was not fully understood. Here we extend our previous observations and show that OT/OTR signals stimulate the Gi protein to interfere with PKA activity to reduce CERB phosphorylation, causing a disruption of its transcription activity for Cav1.2 expression. This action of OT is well comparable to parasympathetic (vagal) neuronal agonist acetylcholine, and adversarial to sympathetic β-adrenergic agonist isoproterenol (Figure 6). According to recent studies, OT and OTR are widely recognized in the heart and vessels [13] in consistent with our study. The OTR can be a couple to distinct G proteins leading to different functional effects [14]. OTR coupling to the Gq/11 protein activates the PLC pathway, which leads to diacylglycerol (DAG)-protein kinase C (PKC) activation and inositol 1,4,5-trisphosphate (IP_3_)-dependent cellular [Ca^2+^]_I_ elevation [14]. These signal pathways have been implicated in processes thought to underlie smooth muscle contraction and NO production. These signal pathways leading to functions that result in a reduction of heart rate and contraction, and vasodilation are reportedly regulated by actions of ANP-guanosine-3′,5′-cyclic monophosphate (cGMP), and NO-cGMP systems [15]. Therefore, molecular signaling of OT in the heart is estimated to be largely dependent on the Gq/11 protein and its downstream targets. The other signaling of OT in the heart which may account for the cardiovascular effect is the pertussis toxin-sensitive GTP-binding protein, Gi [14]. Our results in this study firmly support the action of the Gi protein to modify the expression of the Cav1.2 channel mediated by a transcription factor CREB. The effect of OT on the phosphorylation of CREB in the brain has widely been recognized [14]. OT regulates stress-induced corticotropin releasing factor (CRF) gene transcription via a cAMP/PKA-dependent mechanism through CREB-regulated transcription coactivator 3 [16]. In the heart, I_Ca.L_ reduction was observed in CREB knockout cardiomyocytes along with a reduction of the Cav1.2 encoding mRNA CACNA1C [17]. The obvious question of importance asks where and how CREB interacts with the CACNA1C gene. Target genes of CREB reportedly include consensus sites for CREB-binding on their promotor region, whose molecular structure contains eight-base-pair palindrome (TGACGTCA) or single (CGTCA) motif CREs [18]; there are four CREB-binding sites in the putative promotor region of CACNA1C within −2000 bp from transcription starting site, which is required for activation of transcriptional activity [19,20]. CREB is activated by phosphorylation on Ser−133. Consistent with these observations, CREB could be a candidate factor for CACNA1C transcription downregulation by OT. Taken together, it is postulated that OT plays a crucial role in regulating CREB expression in the heart, which is important for the physiological function of the myocardium, including intracellular Ca^2+^ homeostasis. 

### 3.2. Action of OT on the Cardiovascular System though Gi-Dependent Pathway

OT is a potent uterine stimulant that is usually used for the induction of labor and control of postpartum hemorrhage. Acute application OT (intravenous injection) acts on the smooth muscle of the uterus to stimulate contractions through DAG-PKC activation and IP_3_-dependent cellular [Ca^2+^]_i_ elevation [14]. However, OT does not usually possess cardiovascular effects, such as elevation of blood pressure. Meanwhile, plasma OT concentration in nonpregnant women was <1 μU/mL. The mean plasma OT concentration between weeks 4 and 40 of pregnancy was 66–165 μU/mL [21]. Nevertheless, maternal blood pressure during pregnancy does not correspond to the action of OT through Gq/11-IP_3_-dependent elevation of intracellular Ca^2+^ concentration. Although serum circulating OT is believed to prevent hypertension, and to be cardioprotective by reducing the inflammatory response and improving cardiovascular function [15], signaling pathway of OT through Gq/11-IP_3_-dependent activation of PKC and IP_3_ may not be appropriate enough to illustrate the beneficial effect of OT in the cardiovascular system. In this context, downregulation of the Cav1.2 channel in cardiomyocytes, as shown in this study and possibly in vascular smooth muscle cells, may account for beneficial action of OT in cardioprotection and control of blood pressure during pregnancy. In accordance with these mechanisms, our results open to the understanding of the novel action of OT as a cardiovascular hormone.

### 3.3. Study Limitation

Various limitations of our study should be addressed: (1) This was an experimental study by use of rodent, and the results may not be extrapolated to human subjects, (2) downregulations of phosphorylated-CREB and Cav1.2 by OT were assessed only in cardiomyocytes, but not in vascular smooth muscle cells, (3) actions of OT on other CREB-associated transcription regulators, such as CREB-binding protein (CBP) and CREB-regulated transcriptional coactivators (CRTCs), were not assessed in this study, and (4) possible changes of Ca^2+^ channel β and α2/δ subunits were not examined. These questions should be addressed thoroughly in future studies. In addition, it is important to keep in mind that the concentration of OT in this study (100 nM) was much higher than the serum OT concentration in humans in pregnancy and postpartum (0.5–2.5 nM) [22]. Therefore, it is important to exercise caution when interpreting data from this animal model. 

### 3.4. Conclusions

In the present study, we have successfully demonstrated a novel signal pathway of OT to regulate cardiac L-type Ca^2+^ expression, which is largely associated with cardiac excitability and intracellular Ca^2+^ homeostasis. This pathway is expected to attenuate Ca^2+^ overload in cardiomyocytes and smooth muscle cells when the Gq/IP_3_ pathway is overly activated by excessive OT. With regard to the pharmaceutical aspect, these mechanisms of OT on cardiomyocytes provide the precise pharmacological rationale for the prevention and treatment of cardiovascular diseases based on the Ca^2+^-overload pathophysiological condition of the heart.

## 4. Materials and Methods 

All experimental protocols were approved in advance by the Ethics Review Committee for Animal Experimentation of Oita University School of Medicine (No. C004003, No. G004006), and were carried out according to the guidelines for animal research of the Physiological Society of Japan to minimize the number of animals used, as well as their suffering. 

### 4.1. Animals and Surgery

Adult male Wistar rats (300–400 g) used for this study were obtained from Kyudo Co., Ltd. (Fukuoka, Japan). All rats were housed three per cage in a temperature-controlled room (23 °C) with a constant light/dark cycle (lights on 8:00–20:00 h). Tissues from adult rat atrium or ventricle were dissected in liquid nitrogen and stored at −80 °C until real-time PCR analysis was performed. Neonatal rat ventricular myocytes were prepared from 1- to 2-day-old Wistar rats as described previously [23]. The cardiomyocytes were maintained at 37 °C under 5% CO_2_ in Dulbecco’s modified Eagle’s medium (DMEM) supplemented with 10% fetal bovine serum for 24 h, then the medium was changed to serum-free condition containing oxytocin (10^−8^ M to 10^−6^ M) or oxytocin with other signal inhibitors (see the section on Inhibitor experiments). The spontaneously beating myocytes were subjected to electrophysiological experiments 24 h after treatment with OT.

### 4.2. Chemicals 

OT and OTRA [d(CH_2_)_5_Tyr(Me)^2^,Thr^4^,Tyr-NH_2_^9^]-vasotocin were purchased from Bachem (Bachem Peninsula Laboratories, Inc., Bubendorf, Switzerland). U73122 was obtained from Sigma (St. Louis, MO, USA). Isoproterenol, pertussis toxin, acetylcholine, forskolin, and other chemicals were from Wako Pure Chemical Industries (Osaka, Japan). OT and OTRA were dissolved in 5% acetic acid, while U73122 and forskolin were dissolved in dimethyl sulfoxide (DMSO) as stock solutions (5–20 mM), and then diluted to final concentrations in cell culture solutions. The final concentration of DMSO or acetic acid in the bathing solution was 0.01% or less. Prior to electrophysiological and real-time PCR analyses, these concentrations of DMSO or acetic acid were confirmed not to significantly influence I_Ca.L_ or Cav1.2 mRNA expression. 

### 4.3. Inhibitor Experiments

Several inhibitors were used to identify the role of intracellular signaling pathways in the action of OT. The concentration of these inhibitors used in this study was optimized based on the results of previous studies [13,19,24]: A highly specific oxytocin receptor antagonist, [d(CH_2_)_5_Tyr(Me)^2^,Thr^4^,Tyr-NH_2_^9^]-vasotocin 1 μM, a specific inhibitor of the Gi class pertussis toxin (PTX) 100 ng/mL, a phospholipase C (PLC) inhibitor (U73122) 2 μM, and an adenylate-cyclase activator forskolin (FSK) 10 μM. Each inhibitor was added alone or simultaneously with OT for 24 h to determine L-type Ca^2+^ channel expression and to assess the level of CREB phosphorylation.

### 4.4. Electrophysiological Measurements

Whole-cell voltage clamp experiments were performed as described previously [25]. L-type Ca^2+^ channel current (I_Ca.L_) was recorded from a holding potential (V_H_) of −50 mV followed by various test potentials. I_Ca.L_ density was obtained by normalizing I_Ca.L_ to the cell capacitance. All experiments were conducted at 37 °C. For measuring I_Ca.L_, the bath solution was composed by Na^+^- and K^+^-free solution contained (mM): Tetraethylammonium chloride (TEA-Cl) 120, CsCl 6, 4-Aminopyridine (4-AP) 5, MgCl_2_ 0.5, 4,4P-diisothiocyanostilbene-2,2P-disulfonic acid (DIDS) 0.1, HEPES 10, CaCl_2_ 1.8, and glucose 10 (pH of 7.4 adjusted with TEA-OH). The pipette solution contained (mM): CsCl 130, Mg-ATP 2, EGTA 5, and HEPES 10 (pH of 7.2 adjusted with 1 M CsOH).

### 4.5. Quantitative Real-Time PCR

Total RNA was extracted from rat neonatal ventricular cardiomyocytes and adult hearts using TRIzol (Invitrogen, Carlsbad, CA, USA) 24 h after the treatment with agents described above. The single-stranded cDNA was synthesized from 1 μg of total RNA using Transcriptor First Strand cDNA Synthesis Kit (Roche Molecular System Inc, Alameda, CA, USA). Real-time PCR was performed on Light Cycler (Roche) using the FastStart DNA Master SYBR Green I (Roche) as a detection reagent. The oligonucleotide primers used for real-time quantitative PCR are described in our previous reports [11]. Rat glyceraldehydes-3-phosphate dehydrogenase (GAPDH; GeneBank accession no. M17701) mRNA was used as an internal control. Data were calculated by 2^−ΔΔCT^ and presented as fold change of transcripts for Ca_v_1.2 genes in myocytes and normalized to GAPDH (defined as 1.0 fold). The size of the PCR products was confirmed by 2% agarose gel electrophoresis.

### 4.6. Western Blot Analysis

Cultured neonatal rat cardiomyocytes were treated with OT (10^−7^ M) and OT with other signal inhibitors or agonists for 24–48 h in DMEM. All the subsequent steps were performed at 4 °C. Cardiomyocytes were washed twice with ice-cold PBS, scrapped, and transferred to a centrifuge tube. Cytoplasmic or nuclear proteins from cardiomyocytes were prepared using NE-PER Nuclear and Cytoplasmic Extraction Reagents based on the manufacturer’s instruction (Pierce, Rockford, IL, USA). The extracted protein concentration was determined by the BCA protein assay kit (Pierce). Samples containing 40 μg were denatured at 95 °C for 5 min in loading buffer [Tris-HCl (pH 6.8) 250 mM, 4% SDS, 1% β-mercaptoethanol, 1% bromophenol blue, and 20% glycerol], separated by SDS-polyacrylamide gel electrophoresis using 8% polyacrylamide gel, and then transferred from the gel to a PVDF membrane (Hybond-P; GE Healthcare Bio-Sciences, Piscataway, NJ, USA). To prevent nonspecific binding, the blotted membranes were blocked with 5% skim milk in tris-buffered saline (TBS) with 0.1% Tween 20 (TBST) for 1 h at room temperature and then probed overnight at 4 °C with an anti-CREB antibody (1:1000, Cell Signaling, Beverly, MA, USA) and phosphospecific antibody against anti-pCREB (Ser 133) (1:1000, Cell Signaling). The blot was visualized with anti-rabbit IgG horseradish peroxidase-conjugated secondary antibodies (1:2000, American Qualex, CA, USA) and an ECL plus Western Blotting Detection System (GE Healthcare Bio-Sciences), and the results were exposed on Biomax Light film (Eastman Kodak; Rochester, NY, USA). Band density was measured with Scion image software and background subtraction algorithms. Protein loading was controlled by probing Western blots with anti-β-actin (1:1000, Cell Signaling) and normalizing ion channel protein band intensities to β-actin values.

### 4.7. Protein Kinase A (PKA) Assay

Cultured neonatal rat cardiomyocytes were treated with OT (10^−7^ M) and OT with other signal inhibitors for 24 h in DMEM, and cells then underwent lysis in a commercial lysis buffer containing protease inhibitors and phosphatase inhibitors. A 15-µg quantity of cytoplasmic protein was used for PKA reaction with a PepTag assay kit (ProMega, Madison, WI, USA), following the manufacturer’s protocol. After phosphorylation, the PKA-specific fluorescent peptide substrate kemptide alters the net charge from +1 to −1, and the phosphorylated and non-phosphorylated forms can be separated on agarose gels. Enzyme activity was expressed as substrate catalyzed unit/mg protein. 

### 4.8. Statistical Analysis

Data are expressed as mean ± S.E. Between groups and among groups comparisons were conducted with one-way ANOVA followed by a Scheffe test. A *p* value < 0.05 was considered significant.

## Figures and Tables

**Figure 1 membranes-11-00234-f001:**
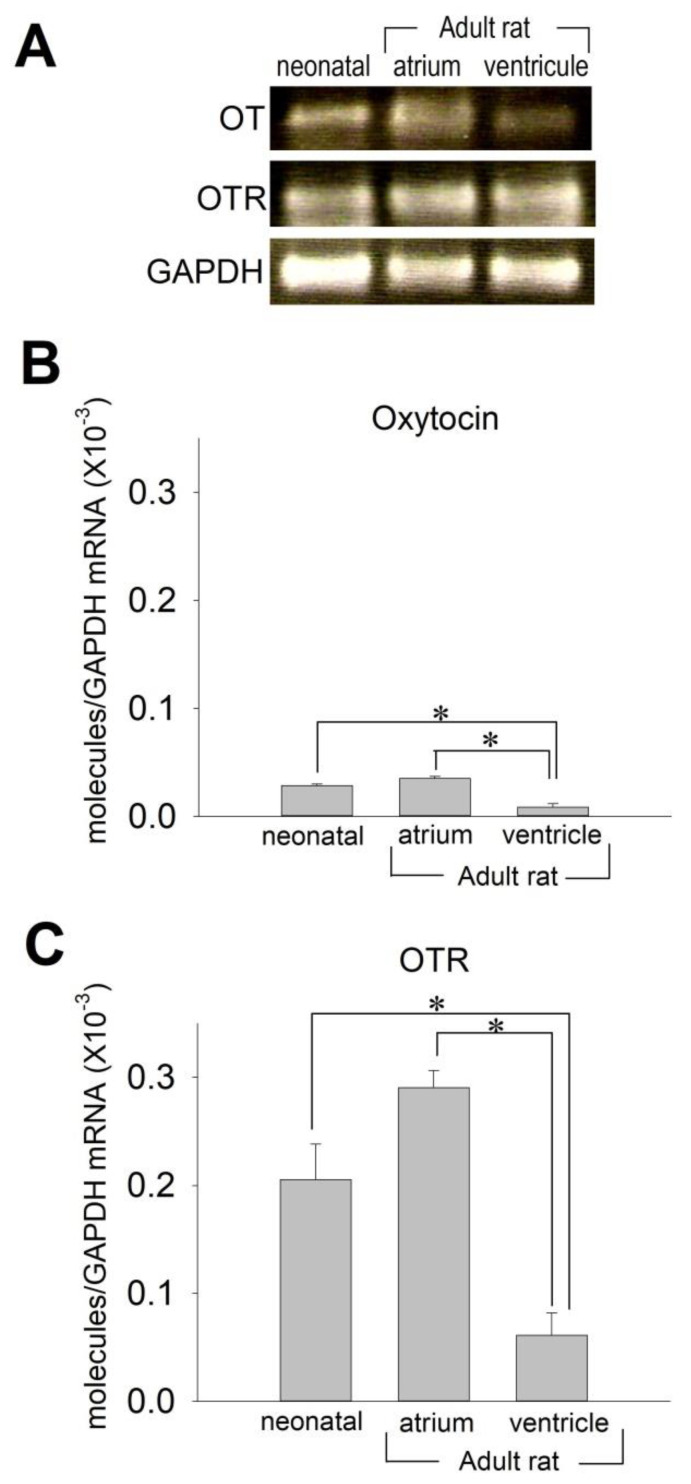
Detection of oxytocin (OT) and OT receptor (OTR) mRNA in adult and neonatal rat hearts. (**A**) The photograph presents OT and OTR RT-PCR products after electrophoresis on 2% agarose in ethidium bromide from the following targets: Neonatal ventricle (left), adult atrium (middle), and adult ventricle (right). (**B**,**C**) Relative quantification of OT (**B**) and OTR (**C**) mRNA in rat neonatal ventricle and adult heart, atrium, and ventricle, by real-time PCR. Changes in expression of OT and OTR normalized to glyceraldehydes-3-phosphate dehydrogenase (GAPDH) were evaluated. Data are expressed as mean ± S.E. (*n* = 8). * *p* < 0.05 between groups indicated.

**Figure 2 membranes-11-00234-f002:**
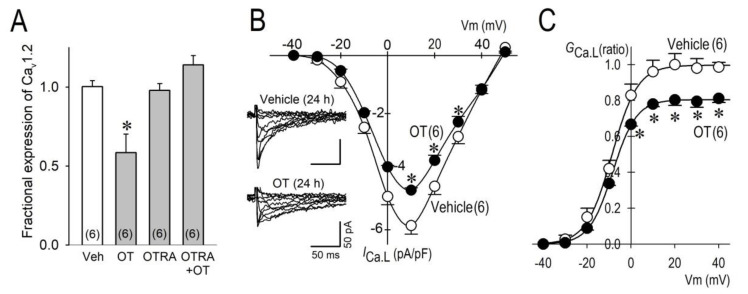
Long-term effects of OT on Cav1.2 expression and I_Ca.L_. (**A**) Expression ratio of Cav1.2 mRNA after treatment with OT (100 nM), OTR antagonist OTRA (1 μM), and OT (100 nM) with OTRA (1 μM) for 24 h. (**B**) Representative I_Ca.L_ traces in the vehicle and OT (inset) applied for 24 h, and their group data of current (I)-voltage (V) relationship. Current traces were obtained from a holding potential of −40 mV to test potentials up to 50 mV with 10 mV increments. (**C**) Group data of chord conductance (G_Ca.L_)-voltage (V) relationship for vehicle and OT constructed from I-V curves in panel (**B**) * *p* < 0.05 between groups at the same potentials. The numbers of cells are indicated in parentheses.

**Figure 3 membranes-11-00234-f003:**
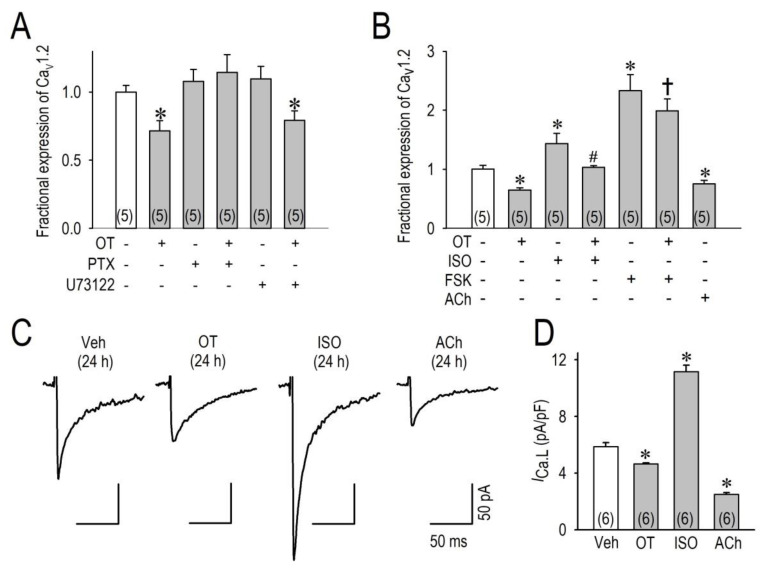
Changes of Cav1.2 mRNA and I_Ca.L_ by OT. (**A**) Regulation of Cav1.2 expression (ratio) by OT (100 nM) was assessed with/without an inhibitor of the Gi subunit of heterotrimeric G proteins pertussis toxin (PTX) of 100 ng/mL or an inhibitor of the phospholipase C pathway (U73122) of 2 μM. (**B**) Regulation of Cav1.2 expression (ratio) by OT was assessed with/without 100 nM isoproterenol (ISO), 10 μM forskolin (FSK) or 1 μM acetylcholine (ACh). (**C**) Typical I_Ca.L_ traces recorded at a test potential of 0 mV from a holding potential of −50 mV in cardiomyocytes cultured with the vehicle, OT, isoproterenol (ISO), or acetylcholine (ACh) at the same concentrations in panel (**B**) for 24 h. (**D**) Group data for I_Ca.L_ recorded at 0 mV in cardiomyocytes treated with OT, isoproterenol (ISO), or acetylcholine (ACh). Data are expressed as mean ± SE. * *p* < 0.05 vs. vehicle, # *p* < 0.05 vs. ISO, † *p* < 0.05 vs. FSK. The numbers of cells are indicated in parentheses.

**Figure 4 membranes-11-00234-f004:**
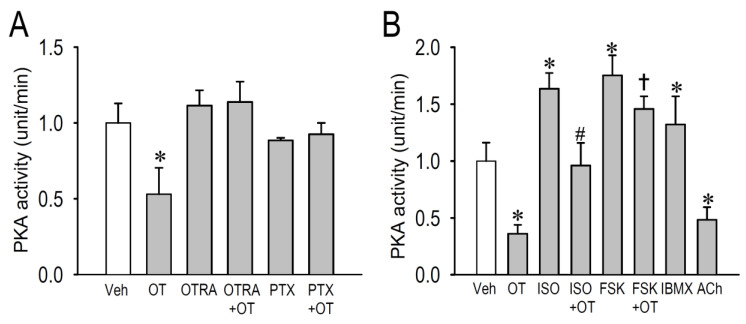
Changes of PKA activity by OT assessed with/without cellular OTR/Gi/cAMP regulators. (**A**,**B**) PKA activity in neonatal cardiomyocytes were assessed after treatment with/without OT (100 nM), OTRA (1 μM), pertussis toxin (PTX, 100 ng/mL), isoproterenol (ISO, 100 nM), forskolin (10 μM), a non-selective inhibitor of phosphodiesterase isobutylmethylxanthine, 1-methyl-3-isobutylxanthine (IBMX, 100 μM) or Acetylcholine (ACh, 1 μM) for 24 h. Data are expressed as mean ± SE (*n* = 6). * *p* < 0.05 vs. vehicle, # *p* < 0.05 vs. ISO, † *p* < 0.05 vs. FSK.

**Figure 5 membranes-11-00234-f005:**
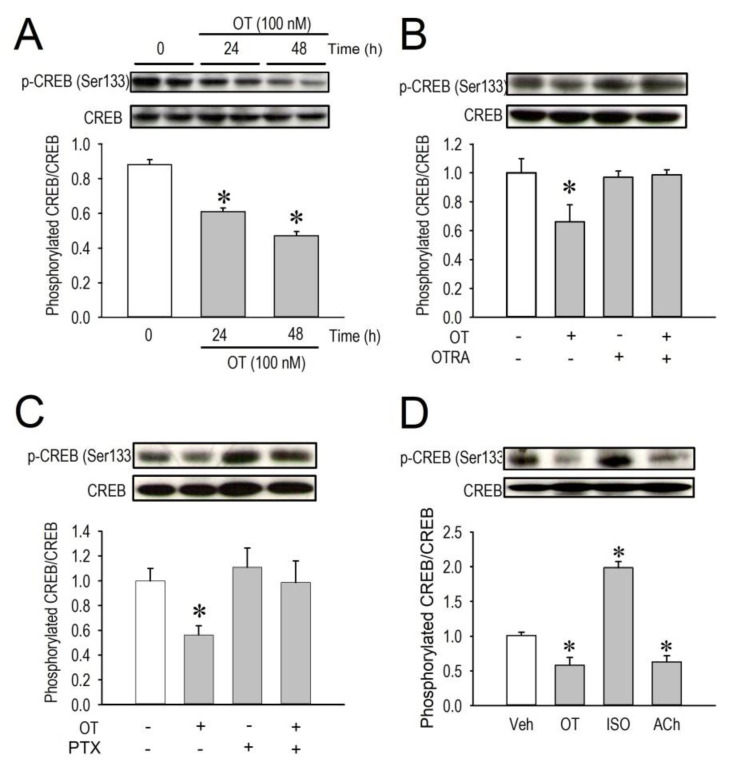
Changes of phosphorylated CREB protein level in the nucleus of cardiomyocytes. Densitometric analysis of phosphorylated-CREB (p-CREB) and CREB protein expression normalized to β-actin. The amount of p-CREB per total CREB proteins (phosphorylated CREB/CREB) was assessed and compared among the groups. (**A**) Time-dependent suppression of p-CREB by OT (100 nM). Upper panel, an immunoblot demonstrating the effect of OT (100 nM, 24 h or 48 h) on total-CREB, p-CREB, and β-actin expression in neonatal rat cardiomyocytes. (**B**,**C**) Inhibitory actions of OTR antagonist (OTRA, 1 μM) and/or pertussis toxin (PTX, 100 ng/mL) on p-CREB. (**D**) Changes of p-CREB by OT (100 nM), isoproterenol (ISO, 100 nM), and acetylcholine (ACh, 10 μM). Data are expressed as mean ± S.E (*n* = 6). * *p* < 0.05, vs. non-treated myocytes (vehicle).

**Figure 6 membranes-11-00234-f006:**
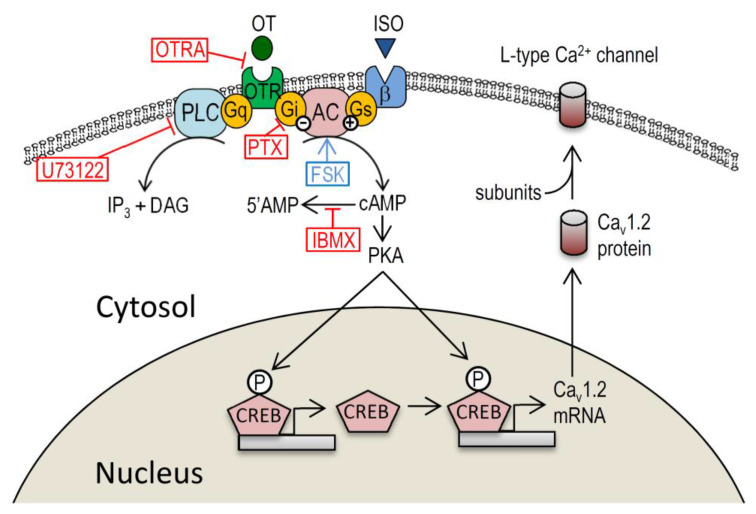
Schematic diagram of hypothetical OT signaling pathways targeting the cytoplasmic membrane and the nucleus in cardiomyocytes. OT-mediated downregulation of the L-type Ca^2+^ channel Cav1.2 expression through a reduction of CREB phosphorylation.

## Data Availability

The data in this study are available from the corresponding author on reasonable request.

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
