# Peer review of "Oxytocin Downregulates the Ca_V_1.2 L-Type Ca^2+^ Channel via Gi/cAMP/PKA/CREB Signaling Pathway in Cardiomyocytes"

_membranes, 2021, doi:10.3390/membranes11040234_

Round 1
Reviewer 1 Report
Morishima et al. clearly show the regulation of ion channels by sex hormone. A female reproductive hormone, oxytocin, downregulates the expression of Cav1.2 channels in rat cardiomyocytes. Oxytocin receptors/Gi/cAMP/PKA/CREB signaling pathway is involved in this downregulating mechanism. This topic is very interesting and important in the research field of ion channels. The majority of experimental data is clear and supports the conclusions in the manuscript. I have some minor comments that may need the authors’ attention to improve the manuscript.
- 1: Is Fig. 1A a representative RT-PCR image for expression change in oxytocin receptors? Expression change of this image does not match the summarized results in Fig. 1C.
- 2: For readers, describe longer exposure effects and concentration dependency of oxytocin on Cav1.2 expression demonstrated in your previous publication (Morishima et al., J Arrhyth, 2010) in the Results or Discussion section.
- 2A, legend: “Functional expression ratio of …”. Remove “Functional”.
- 2A/B: Explain the difference in effects by oxytocin between the downregulation of Cav1.2 (41%) and the decrease of VDCC currents (20.8%).
- Discussion, line 268: “… by the long-term effect, but not by the short-term effect [11]”. Describe specifically; e.g., “… by the long-term effect (24-72 h), but not by the short-term effect (5 min)”.
- Discussion, line 315: Did the authors examine the same experiments using cell lines from human cardiomyocytes?
Author Response
To reviewer #1
Thank you for your very careful reading of our manuscript, and your favorable suggestions on our study. In this revision, we have tried to address each point, and we feel that the manuscript has been improved because of your valuable comments. We hope that these changes in manuscript are satisfactory and that the revised version is acceptable for publication in Membranes. Changes/additional text are yellow-highlighted for clarity.
- Is Fig. 1A a representative RT-PCR image for expression change in oxytocin receptors? Expression change of this image does not match the summarized results in Fig. 1C.
We apologize for demonstrating the photograph of PCR product somewhat inappropriate at a glance in Figure 1A. Because neonate heart contains only a little of fibrotic matrix, total cardiomyocyte per heart weight is higher, and therefore the amount of net cardiomyocytes including GAPDH was higher per heart weight in comparison with adult heart. That is why neonate OT or OTR versus GAPDH ratio does not look like what shown in panel B and C. Because this photo is the only graphical images we obtained, please accept our apology that GAPDH image of neonate heart is brighter than that of adult atrium and adult ventricle.
- For readers, describe longer exposure effects and concentration dependency of oxytocin on Cav1.2 expression demonstrated in your previous publication (Morishima et al., J Arrhyth, 2010) in the Results or Discussion section.
Thank you for your suggestion. We agree with you. It seems to be a good idea to introduce the result of our previous study which triggered us to further explore the mechanism of OT for the Ca2+ channel regulation in this article. In response to your suggestion, we describe the summary of the result in the reference #11 in the discussion section (Page 8 lines 280-285).
- 2A, legend: “Functional expression ratio of …”. Remove “Functional”.
Thank you for picking this up. The word removed.
- 2A/B: Explain the difference in effects by oxytocin between the downregulation of Cav1.2 (41%) and the decrease of VDCC currents (20.8%).
According to our experiment on the Nav1.5 channel (Br J Pharm 157; 404-414, 2009) and the Cav1.2 channel (unpublished data), life-spans of the trans-membrane ion channels are approximately 12-16 hours as a form of the functioning protein at the plasma membrane. Therefore the steady state of the channel population after the stimulation signals applied to the plasma membrane could be estimated to need more than 2-3 times of the life-span periods which include the transcription, translation, and the trafficking process to the plasma membranes. In other words, 24 hours after the onset of new exogenous stimulation to modify the population of ion channels may be too short to observe the steady state of the channel population by patch clamp method. However, when we observed the changes of ICa.L after 48-72 hours application of OXT (J. Arrhyth. 2010, 26, 111-118.), the reduction ratio of ICa.L was nearly identical to that of 24 hours for an unknown reason. The degradation process of the channel protein may be modified by OXT-dependent signal pathway, although we do not have enough data to support this hypothesis. In short, we could not explain the difference between the reduction rate of mRNA and ICa.L by OXT (24 h). We apologize for the shortness of our understanding on the data evaluation.
- Discussion, line 268: “… by the long-term effect, but not by the short-term effect [11]”. Describe specifically; e.g., “… by the long-term effect (24-72 h), but not by the short-term effect (5 min)”.
The part is now changed to the current form as per your request. (Page 8 lines 280-285)
- Discussion, line 315: Did the authors examine the same experiments using cell lines from human cardiomyocytes?
No, we have not examined human cell lines. Because of this, in this revision, we change the statement in a modest way to explain the possible beneficial action of OXT in the text.
Reviewer 2 Report
The manuscript titled “Oxytocin down-regulates the CaV1.2 L-type Ca2+ channel via 2 cAMP/PKA/CREB signaling pathway in cardiomyocytes” submitted by Moroshima et al. describes the influence of long term exposure of cardiomyocytes to oxytocin.
They find a decrease in the L-type channel mediated current levels, which they attribute to a change in channel expression.
While the manuscript is carefully prepared and well written there are several points that need further clarification:
General points:
The authors use a concentration of oxytocin of 100 nM throughout the manuscript. Given the reported KD of 0.3 nM (e.g. Jasper et al. 1995) this is extremely high. This is even more worrisome, if one considers reported plasma oxytocin levels range from pM to low nM. Please comment on this and at least discuss it in the manuscript.
The focus lies on Cav1.x expression changes, however only mRNA changes are reported. These changes need not reflect changes in protein levels. I would thus strongly recommend to use for examples WB to measure expression levels in order to demonstrate that the changes in mRNA are indeed relevant.
Specific points:
Section 2.1 line 12 /Figure 1:
The authors report a “abundant” expression of OT and OTR. They however only show normalized data, so no copy count information can be extracted and thus the significance of these findings is hard to judge. I would strongly suggest to set these data into context by showing performing the same experiment for beta adrenergic and muscarinic receptors.
Section 2.2 sentence 15:
The authors report they have measured T-type channels, but they do not present the data. Please add the T-type data.
Section 2.3 / Figure 3 & 4:
In figure 3 the authors demonstrate, that after 24h treatment with OT or ACh, L-type currents are smaller and that 24 h treatment with Iso leads to larger currents. In figure 4 they demonstrate that this is in line with a decreased / increased PKA activity. They attribute this to a reduced / increased expression of Cav1.2. Given the fact, that PKA phosphorylation is the major regulator of Cav1.2 channel activity, the question arises if this finding could just be an effect of reduced / increased channel phosphorylation and not be due to expression level changes. Thus, I would strongly recommend to use an alternative method such as (nonstationary) noise analysis to measure the channel number and biophysical parameters of Cav1.x channels under different treatment conditions. In addition the authors should check the phosphorylation status of Cav1.x channels under the different treatment conditions.
Author Response
To Reviewer #2
Thank you for your careful reading of our manuscript, and your very important questions, which need to be clarified. In this revision, we have tried to address each point, and we feel that the manuscript has been improved because of your valuable comments. We hope that these changes in manuscript are satisfactory and that the revised version is acceptable for publication in Membranes. Changes/additional text are yellow-highlighted for clarity.
General points:
The authors use a concentration of oxytocin of 100 nM throughout the manuscript. Given the reported KD of 0.3 nM (e.g. Jasper et al. 1995) this is extremely high. This is even more worrisome, if one considers reported plasma oxytocin levels range from pM to low nM. Please comment on this and at least discuss it in the manuscript.
Thank you for your very important comment on the concentration of OT in this investigation. We fully agree with you on this point. Based on recent literatures, we newly make some description on the serum OT concentration in human in pregnancy and the postpartum in the discussion section, and make a warning comment that the concentration of OT in this study (100 nM) is far high in comparison with the physiological concentration range of OT. (Page 9 lines 347-351)
The focus lies on Cav1.x expression changes, however only mRNA changes are reported. These changes need not reflect changes in protein levels. I would thus strongly recommend to use for examples WB to measure expression levels in order to demonstrate that the changes in mRNA are indeed relevant.
We believe, and probably almost all the electrophysiologist as well, that the current density of the Ca2+ channel firmly represent the amount of Ca2+ channel protein at the plasma membrane level. In this sense, the “working” proteins. Because WB only represents the amount of whole proteins of the channel, working or semi-degraded or even in the cytoplasmic ones, functional amount of the channel protein could not be detected by WB. Because of this, we strongly believe that patch clamp analysis for the maximum current of ICa.L is the best methods to measure the functional amount of channel proteins. WB is particularly useful, for instance, to study the channel degradation and internalization process. However, in the current study, we believe that patch-clamp method to evaluate the maximum ICa.L and the maximum GCa.L is the best way to detect the changes of Ca2+ channel protein caused by OT. In addition, we have only 10 days to revise this manuscript, which is too short to make WB. So we cordially ask for your generous understanding of these results.
Specific points:
Section 2.1 line 12 /Figure 1:
The authors report a “abundant” expression of OT and OTR. They however only show normalized data, so no copy count information can be extracted and thus the significance of these findings is hard to judge. I would strongly suggest to set these data into context by showing performing the same experiment for beta adrenergic and muscarinic receptors.
We agree with you on this point again. We intended to look into the rationale based on the experiment in Figure 1 with which it has been used by means of neonatal ventricular cardiomyocytes for the evaluation of OT action in patch clamp analysis and others in this study. For this purpose, some descriptions on the results in Figure 1 are changed. (Page 3 Lines 42-43)
Section 2.2 sentence 15:
The authors report they have measured T-type channels, but they do not present the data. Please add the T-type data.
Thank you for your question on the T-type Ca2+ channel. As you may know, nobody has successfully demonstrated experiments by use of primary cultured nodal cells, sinus node and atrioventricular node, for prolonged cultured periods for 24 hours or so to study electrophysiological feature of cardiomyocytes. Because of this reason, many people use neonatal ventricular cardiomyocytes. We could not detect T-type Ca2+ channel current (ICa.T) in the normal adult ventricular cardiomyocytes, and the current is very small in neonatal ventricular cardiomyocytes. We have successfully demonstrated ICa.T in previous publications, however, ICa.T density is small (~2 pA/pF), thus low signal/noise ratio prevent us to demonstrate good current traces in the Figure, although ICa.T amplitudes are measurable and assessable. That is why we are not willing to show ICa.T traces in the figure. Also this study was not aimed to investigate changes of ICa.T, traces of ICa.T may not be very important to make a conclusion in this study. Because of these reasons, we decided to delete the statement on ICa.T in the text. We hope you accept our decision.
Section 2.3 / Figure 3 & 4:
In figure 3 the authors demonstrate, that after 24h treatment with OT or ACh, L-type currents are smaller and that 24 h treatment with Iso leads to larger currents. In figure 4 they demonstrate that this is in line with a decreased / increased PKA activity. They attribute this to a reduced / increased expression of Cav1.2. Given the fact, that PKA phosphorylation is the major regulator of Cav1.2 channel activity, the question arises if this finding could just be an effect of reduced / increased channel phosphorylation and not be due to expression level changes. Thus, I would strongly recommend to use an alternative method such as (nonstationary) noise analysis to measure the channel number and biophysical parameters of Cav1.x channels under different treatment conditions. In addition the authors should check the phosphorylation status of Cav1.x channels under the different treatment conditions.
We agree with you that your question and comments are very reasonable for this kind of investigation. A set of data including channel mRNA, channel protein and IxL must be the most reliable and convincing way of demonstration for ion channel studies. Because our laboratory had worked for just electrophysiology for many years, and because anti-bodies for channel proteins are expensive, we have not tried experiments for Cav1.2 protein evaluation so far in this OT experiment. Since mRNA reduction by OT has confirmed many times here in this study and our previous publication #11 accompanied by ICa.L reduction, we conclude that protein expression should be reduced by a treatment with OT. And because the revision submission due is within only 7 days after editorial comments arrived, we may not have enough time to perform additional experiments according to your suggestions. Because Cav1.2 channel modulation by OT is a very important study for us, we would like to continue the study further exploring the CREB-associated molecules in this signal pathway. For this purpose, we would like to study membrane and cytosol protein levels of Cav1.2, and phosphorylation levels as well. We would like to ask for your understanding.
Reviewer 3 Report
The paper by Masaki Morishima et al. demonstrates the signaling pathway through which oxytocin (OT) down-regulates Cav1.2 L-type calcium channel expression in neonatal rat ventricular cardiomyocytes. Previously, they described oxytocin down-regulated specifically Cav1.2 L-type calcium channel expression but oxytocin did not change the expression of other important voltage calcium channel in cardiomyocytes (Cav1.3, Cav3.2 or Cav3.1). It is known OT binds the OT receptor (OTR), which belongs to G-protein-coupled receptor (GPCRs) and it can signal via Gq and Gi proteins. Using a specific inhibitor of Gi protein (pertussis toxin), authors demonstrated that OT decreased the Cav1.2 mRNA expression and ICaL via Gi protein. They verified that OT reduced basal PKA activity through the OTR/Gi protein pathway. It is known that PKA can phosphorylate and activate the transcription factor CREB. Masaki Morishima et al. demonstrated that CREB phosphorylation is decreased in presence of OT and this effect was completely blocked by OTRA and pertussis toxin, suggesting again that OT produced its effects though OT/OTR/Gi pathways. They suppose inactivation of PKA activity by OT reduces phosphorylation levels of CREB suggesting a PKA-dependent CREB phosphorylation.
In summary, authors indicate a novel signaling pathways (cAMP/PKA/CREB) through which oxytocin down-regulates the Cav1.2 L-type Ca2+ channel in cardiomyocytes.
The experiments are good executed and organized for its interpretation. However, there are questions and concerns that could improve the present work.
Major concerns:
- Authors show I/V curve of ICaL in Figure 2B and chord conductance (GCa.L)-voltage (V) relationship in Figure 2C. I would like to know if the authors have analyzed the activation-inactivation curve of ICaL to analyze if the properties of the channels could be modified in presence of OT vs. vehicle.
- In the Results section, authors suggest CREB binds to the promoter region of CACNA1C (Cav1.2). It could be interesting to perform a bioinformatic analysis to show possible CREB biding sites in the promoter region of CACNA1C and to add some information about that in the results section.
Minor concerns:
- Line 13-14 - the sentence is not clear; please, could you modify it?
- Line 31-32 -the sentence: …OT plays a role in regulating cardiovascular function. It would be better to say OT plays a role in the regulation of cardiovascular function.
- Line 96 – please, be careful with the spaces and the format.
- Line 112-113 – the sentence: we investigated intracellular molecular mechanism for L-type Ca2+ channel expression by OT. It would be better to say we investigated intracellular molecular mechanism for L-type Ca2+ channel expression regulation by OT.
- Line 121-122, line 136 - It should be clear the cell type that were used in the experiments, were the experiments performed in neonatal rat ventricular cardiomyocytes? Please, mark it in the text.
- Line 147 - please, mark the cell type (neonatal rat ventricular cardiomyocytes)
- Line 155 – This sentence is not correct: These electrophysiological results verified the reduction of Cav1.2 in Figure 2A and our previous observations. Please, change the sentence to “These electrophysiological results verified the reduction of Cav1.2 by OT (Figure 2A) and our previous observations”
- Line 184 - please, delete forskolin, figure 3C and 3D show ICaL in cardiomyocytes after OT, acetylcholine and isoproterenol treatment. Forskolin effect is not show in the figure 3C and D.
- In line 197-198 sentence says the same that sentence of line 203-205. I think it is better to unify the data of vehicle and OT to have the same percentage of reduction in both graphics.
- In the Results section, please different sections of figures must be reference:
- Line 238 – it should be incorporated “(Figure 5B)” after OTRA
- Line 241 – it should be incorporated “(Figure 5C)” after pertussis toxin
- Line 251-253 the sentence is not clear; please, modify it.
- Line 349 – Authors comment: The spontaneously beating myocytes were subjected to electrophysiological experiments 24 h after isolation. I think isolation should be modified by treatment. The experiments were performed 24 h after treatment with OT.
- Figure 1 shows mRNA expression in neonatal ventricular cardiomyocytes and in adult rat heart. In the Materials and Methods section, 4.5. Quantitative real-time PCR please, add adult heart to the sentence: The RNA was extracted from neonatal ventricular myocytes and adult rat heart using TRIzol. (line 381).
- Line 389 – please, delete Cav1.3. Authors have studied the mRNA expression of Cav1.2 but not of Cav1.3.
- In the Materials and Methods section, 4.7. Protein kinase A (PKA) assay. Authors represent PKA activity after 24h of treatment with OT in Figure 4. Please, delete 48h in line 417.
FIGURES:
FIGURE 1:
- In the legend, It is not clear if the experiments were performed in neonatal rat heart or neonatal ventricle, please clear it.
- I think it is not necessary the agarose gel Image.
- In Figure 1B, the Y axis scale should be changed. The differences are not appreciated.
- In Figure 1C, significance should have the same format than Figure 1B (asterisk above the line).
FIGURE 2:
- In the legend, authors should show the holding potential which experiments were performed.
- In figure 2C, OT and vehicle are changed, please modify it: dark circles represent OT and white circles represent vehicle.
FIGURE 3:
- In figure 3C, could you change representative ICaL trace in presence of OT? This trace does not present the classical IcaL kinetic.
FIGURE 4:
- In Results section, authors do not discuss the action of IBMX. Please, remove these data of the Figure 4B or add in the text some information about IBMX results.
- In the legend, please, delete the sentence: Because assay kit enzymes activities vary upon product lot numbers. I think this sentence should be commented in Discussion section or in results.
FIGURE 5:
- In Figure 5A, b-actin image is not necessary. If authors would like to show it, please, change it for a better image. It is difficult to appreciate the different lanes.
- In figure 5D, please, change CREB image for a better image. It is difficult to appreciate the different lanes.
- In the legend, it should be marked the time of treatment in panel B, C and D.
FIGURE 6:
- Authors should comment IBMX effect in the text or remove it in the image.

Author Response
To Reviewer #3
Thank you for your very careful reading of our manuscript, and your very valuable comments on our study. Though my academic years of roughly 30 years, I have never experienced such enthusiastic comments and sparing no effort to help others study like this letter. We are greatly obliged for your informative comments throughout the manuscript. I have even intense wishes to discuss science with you personally. Again, we are very happy to have your criticism over our study. We change and modify the manuscript nearly completely along your comments and suggestions in this revision, and we feel that the manuscript has been improved because of your valuable thoughtful guidance. We hope that these changes in manuscript are satisfactory and that the revised version is acceptable for publication in Membranes. If any, we are not opposed to any editorial changes you may make. Changes/additional text are yellow-highlighted for clarity.
Major concerns:
Authors show I/V curve of ICaL in Figure 2B and chord conductance (GCa.L)-voltage (V) relationship in Figure 2C. I would like to know if the authors have analyzed the activation-inactivation curve of ICaL to analyze if the properties of the channels could be modified in presence of OT vs. vehicle. In the Results section, authors suggest CREB binds to the promoter region of CACNA1C (Cav1.2). It could be interesting to perform a bioinformatic analysis to show possible CREB biding sites in the promoter region of CACNA1C and to add some information about that in the results section.
Thank you for your interest on ICa.L gating properties after OT treatment. Although we have worked many years for ion channel gating study, we have not explored the steady-state inactivation properties of ICa.L in this study. This is because the IV curve and the GCa.L curve were not shifted by OT treatment. Also because ICa.L steady-state inactivation curve is not a trivial one, which is also influenced by a degree of voltage-dependent facilitation property, we did not further examined it. We would like to ask for your understanding. While the question on the potential target site of CREB must be very important to understand the insight of this signal pathway. We also wish to know the mechanism for this. Thus in response to your request, we tried to perform bioinformatic analysis for CREB action in reference to previous publications. Here is the result.
Target genes of CREB reportedly include consensus sites for CREB binding on their promotor region, whose molecular structure contains eight-base-pair palindrome (TGACGTCA) or single (CGTCA) motif CREs; there are four CREB binding sites in putative promotor region of CACNA1C within -2000bp from transcription starting site, which is required for activation of transcriptional activity (Tsai et al., Circ Res 2007;100:1476-1485, Yang et al., Heart Rhythm 2018;15:741-749). CREB is activated by phosphorylation on Ser-133. Consistent with these observations, CREB could be a candidate factor for CACNA1C transcription down-regulation by OT. (Page 8 Lines 309-316)
Minor concerns:
Line 13-14 - the sentence is not clear; please, could you modify it?
We have changed some words to make the meaning clearer as follows.
Here, we investigated molecular mechanism of the action of OT on cardiomyocyte membrane excitation focusing on the L-type Ca2+ channel.
Line 31-32 -the sentence: …OT plays a role in regulating cardiovascular function. It would be better to say OT plays a role in the regulation of cardiovascular function.
We have changed the words according to your suggestion. - that OT plays a role in the regulation of cardiovascular function.
Line 96 – please, be careful with the spaces and the format.
“cAMP- dependent” is replaced by “cAMP-dependent”; a space between – and dependent was removed.
Line 112-113 – the sentence: we investigated intracellular molecular mechanism for L-type Ca2+ channel expression by OT. It would be better to say we investigated intracellular molecular mechanism for L-type Ca2+ channel expression regulation by OT.
The sentence was changed to what you suggested.
Line 121-122, line 136 - It should be clear the cell type that were used in the experiments, were the experiments performed in neonatal rat ventricular cardiomyocytes? Please, mark it in the text.
The word “ventricular” was inserted in the line 123.
Line 147 - please, mark the cell type (neonatal rat ventricular cardiomyocytes)
“neonatal rat ventricular” inserted in the line 149.
Line 155 – This sentence is not correct: These electrophysiological results verified the reduction of Cav1.2 in Figure 2A and our previous observations. Please, change the sentence to “These electrophysiological results verified the reduction of Cav1.2 by OT (Figure 2A) and our previous observations”
The sentence was corrected as suggested.
Line 184 - please, delete forskolin, figure 3C and 3D show ICaL in cardiomyocytes after OT, acetylcholine and isoproterenol treatment. Forskolin effect is not show in the figure 3C and D.
“forskolin” deleted.
In line 197-198 sentence says the same that sentence of line 203-205. I think it is better to unify the data of vehicle and OT to have the same percentage of reduction in both graphics.
Thank you for your consideration and suggestions for the way of data presentation. We partially agree with you that it is convincing and straight forward if we combine them together. However, as we describe in the figure legend in Figure 4, measurements of PKA activity by use of the assay kits depend on their enzyme activity upon their product lot number. Because of this, data in Figure 4A (OTRA, PTX, etc.) and Figure 4B (ISO, FSK, etc.) are hardly comparable even after the correction of OT value against vehicle at each panel. One of the important messages in this figure include that a reduction of PKA activity by OT in Figure 4A was confirmed by a different enzyme kit in Figure 4B. Also we believe that reduction rates of PKA activity by OT in Figure 4A (by 47%) and Figure 4B (by 64%) are less important than the fact that OT qualitatively decreased the PKA activity as shown by two independent assay kits. Because of these reasons, we would like to ask for your understanding on the way of data presentation as shown in this form.
In the Results section, please different sections of figures must be reference:
Line 238 – it should be incorporated “(Figure 5B)” after OTRA
(Figure 5B) inserted.
Line 241 – it should be incorporated “(Figure 5C)” after pertussis toxin
(Figure 5C) inserted.
Line 251-253 the sentence is not clear; please, modify it.
We have reworded the sentence as follows.
Our study demonstrates that OT down-regulate the Cav1.2 channel leading to a reduction of ICa.L in cardiomyocytes as a long-term effect, which is accompanied by a decrease of p-CREB in the nucleus.
Line 349 – Authors comment: The spontaneously beating myocytes were subjected to electrophysiological experiments 24 h after isolation. I think isolation should be modified by treatment. The experiments were performed 24 h after treatment with OT.
The method was changed to what you suggested.
“The spontaneously beating myocytes were subjected to electrophysiological experiments 24 h after treatment with OT.”
Figure 1 shows mRNA expression in neonatal ventricular cardiomyocytes and in adult rat heart. In the Materials and Methods section, 4.5. Quantitative real-time PCR please, add adult heart to the sentence: The RNA was extracted from neonatal ventricular myocytes and adult rat heart using TRIzol. (line 381).
The sentence was corrected to what you suggested.
Total RNA was extracted from rat neonatal ventricular cardiomyocytes and adult heart using TRIzol
Line 389 – please, delete Cav1.3. Authors have studied the mRNA expression of Cav1.2 but not of Cav1.3.
Cav1.3, deleted.
In the Materials and Methods section, 4.7. Protein kinase A (PKA) assay. Authors represent PKA activity after 24h of treatment with OT in Figure 4. Please, delete 48h in line 417.
“48 h” deleted.
FIGURES:
FIGURE 1:
In the legend, It is not clear if the experiments were performed in neonatal rat heart or neonatal ventricle, please clear it.
I think it is not necessary the agarose gel Image.
I response to your request, we have clarified the description as shown by yellow highlight in the text. Although you say the agarose gel images are not necessary, it is also true that the images are more convincing than the graph. We also believe that this gel image is helpful to understand the existence of OT and OTR in the heart. We would like to ask for your understanding to have this image presented in the figure.
In Figure 1B, the Y axis scale should be changed. The differences are not appreciated.
We understand your comments that the expanded Y scale helps us to recognize the difference among the three. However, this scale is helpful for us to measure the difference of expression between oxytocin (OT) and OTR at a glance. Namely, OTR expressions are much larger than those of OT in the heart. Because of this reason, we would like to show the expression OT in this scale range. We would like to ask for your understanding.
In Figure 1C, significance should have the same format than Figure 1B (asterisk above the line).
The figure symbol is changed in position.
FIGURE 2:
In the legend, authors should show the holding potential which experiments were performed.
Current traces were obtained from a holding potential of -40 mV to test potentials up to 50 mV with 10 mV increments.
In figure 2C, OT and vehicle are changed, please modify it: dark circles represent OT and white circles represent vehicle.
We are sorry for this bad mistake in Figure 2C. Open circle for Vehicle and filled circle for OT. Thank you for your careful inspection.
Labels “OT” and “Vehicle” are changed in position.
FIGURE 3:
In figure 3C, could you change representative ICaL trace in presence of OT? This trace does not present the classical IcaL kinetic.
In response to your request, we have replaced a new current trace for OT in the revision.
FIGURE 4:
In Results section, authors do not discuss the action of IBMX. Please, remove these data of the Figure 4B or add in the text some information about IBMX results.
In the text, we added a description on IBMX as follows as per your request.
“,and a non-selective phosphodiesterase inhibitor 3-isobutyl-1-methylxanthine (IBMX) in the culture medium to confirm their action to increase PKA activity in cardiomyocytes as positive controls.”
In the legend, please, delete the sentence: Because assay kit enzymes activities vary upon product lot numbers. I think this sentence should be commented in Discussion section or in results.
In response to your request, we have moved the sentence to the RESULT section.
FIGURE 5:
In Figure 5A, b-actin image is not necessary. If authors would like to show it, please, change it for a better image. It is difficult to appreciate the different lanes.
In response to your request, we removed b-actin image from Figure 5A.
In figure 5D, please, change CREB image for a better image. It is difficult to appreciate the different lanes.
We are sorry that CREB image in this panel 5D is the best image we have. Although this picture does not look great, we would like to ask you and the potential readers of the article to accept this as it is.
In the legend, it should be marked the time of treatment in panel B, C and D.
FIGURE 6:
Authors should comment IBMX effect in the text or remove it in the image.
Result section in the revision, we have added some words describing the effect of IBMX on PK activity in Figure 4A.
Round 2
Reviewer 3 Report
Thank you for your efforts to answer and modify all the changes suggested. The changes in the manuscript are satisfactory and have improved it.